# Modulation of Hedgehog Signaling for the Treatment of Basal Cell Carcinoma and the Development of Preclinical Models

**DOI:** 10.3390/biomedicines10102376

**Published:** 2022-09-23

**Authors:** Meghan W. Dukes, Thomas J. Meade

**Affiliations:** Departments of Chemistry, Molecular Biosciences, Neurobiology, and Radiology Northwestern University, Evanston, IL 60208, USA

**Keywords:** basal cell carcinoma, HEDGEHOG signaling, smoothened inhibitors, Gli inhibitors, pre-clinical models

## Abstract

Basal Cell Carcinoma (BCC) is the most commonly diagnosed cancer worldwide. While the survivability of BCC is high, many patients are excluded from clinically available treatments due to health risks or personal choice. Further, patients with advanced or metastatic disease have severely limited treatment options. The dysregulation of the Hedgehog (Hh) signaling cascade drives onset and progression of BCC. As such, the modulation of this pathway has driven advancements in BCC research. In this review, we focus firstly on inhibitors that target the Hh pathway as chemotherapeutics against BCC. Two therapies targeting Hh signaling have been made clinically available for BCC patients, but these treatments suffer from limited initial efficacy and a high rate of chemoresistant tumor recurrence. Herein, we describe more recent developments of chemical scaffolds that have been designed to hopefully improve upon the available therapeutics. We secondly discuss the history and recent efforts involving modulation of the Hh genome as a method of producing in vivo models of BCC for preclinical research. While there are many advancements left to be made towards improving patient outcomes with BCC, it is clear that targeting the Hh pathway will remain at the forefront of research efforts in designing more effective chemotherapeutics as well as relevant preclinical models.

## 1. Introduction

Keratinocyte cancers, or nonmelanoma skin cancers (NMSCs), are the most commonly diagnosed cancers worldwide [1,2]. In the United States alone, one in every three to five Caucasian people are expected to develop an NMSC in their lifetime, with estimates as high as 4 million cases diagnosed each year [3,4,5]. Approximately 80% of all NMSCs are characterized as basal cell carcinomas (BCC), where uncontrolled growth of the basal cell population of the epidermis leads to tumorigenesis (Figure 1) [6,7]. The overwhelming number of BCC diagnoses requires ample research and medical attention for the development of effective treatment and prevention strategies. 

Gorlin Syndrome (GS) is a rare autosomal dominant disease comprising a small percentage of the BCC community and approximately 0.05% of the population [8]. In total, 90% of patients with GS experience the uncontrolled growth of multiple BCCs alongside various developmental abnormalities including those associated with holoprosencephaly and malignant medulloblastomas [8,9]. Sporadic BCC accounts for the predominant population of BCC patients. The primary cause of sporadic BCC is prolonged exposure to ultraviolet (UV) radiation from the sun [10,11]. The risk of developing BCC increases with light-skin pigmentation, age, and sunburn frequency during youth [12]. Other risk factors include family history of melanoma, blonde/red hair phenotype, and men are more susceptible to BCC than women [12,13].

A significant burden to the BCC patient community is the high rate of recurrence. BCC is clinically designated as low or high risk depending on the likelihood of recurrence [14]. However, BCC more commonly recurs in an entirely different location on the body. For primary tumor locations, the recurrence rate depends heavily on the method of treatment (discussed in depth in Section 2), with a range from 1 to 70% after 5 years [15,16]. Larger tumors also experience an increased likelihood of relapse [16]. More strikingly, the three-year risk of developing a second BCC lesion is estimated between 41–44% [15,17,18,19], and the likelihood increases with each additional lesion. Once diagnosed, approximately 50% of patients will battle BCC again.

BCC is classified in three primary identities: superficial (10–30%), nodular (60–80%), and morpheaform/infiltrative (<10%) [7,20]. Each differ in physical and histopathological behavior and exhibit differential relapse rates [21]. Superficial and nodular BCCs are less likely to recur, whereas infiltrative BCCs are more challenging to treat permanently [22]. Additionally, the different subtypes have variable occurrence rates on different skin areas. Nodular BCC is most commonly found on the face, while superficial BCC affects the torso and hands more frequently [23]. Infiltrative BCC is the most aggressive form and can often lead to the destruction of nearby healthy tissue [24]. Due to these differences in risk classification and behavior, selective care must be taken when deciding which treatment option to pursue for an individual BCC patient.

BCC’s overall survivability is very high, with estimates for mortality being less than 0.5% [25,26]. However, the exceptionally high number of BCC diagnoses means that even a low mortality rate produces significant BCC-related cancer deaths [5]. The American Cancer Society estimates this population at around 2000 NMSC-related deaths annually, primarily attributed to metastatic BCC complications.

With ever-increasing rate of BCC diagnoses, vast differences in subtypes, and extremely high rate of recurrence, it is imperative to focus BCC research efforts on preventing disease and improving disease outcomes. In this review, we outline the current state of research progress towards improving BCC treatment with a focus on the molecular drivers of BCC pathology. We first outline the treatment strategies employed in the clinic for BCC and highlight their advantages, disadvantages, and any known patient restrictions. Secondly, we elaborate on the Hedgehog (Hh) signaling cascade, the molecular driver of BCC. Thirdly, we provide an in-depth review of molecular chemotherapeutics that target the Hh pathway and how genetic modulation of Hh regulators has been used to develop in vivo BCC models. Finally, we provide a prospective on the state of the field and present opinions on future research priorities.

## 2. Predominant Treatment Options for BCC

Treatment strategies for BCC vary by subtype of the disease, size of the lesion, and patient age and preference. While the following list does not represent every option for BCC therapy, several predominant options are discussed in this section. Table 1 outlines the advantages and disadvantages of each, including any imperative patient restrictions.

### 2.1. Surgical Resection

Surgical intervention is by far the preferred treatment option for BCC owing to the highest rate of complete tumor clearance and lowest recurrence rates [15]. Two common forms of surgical intervention are wide local excision (WLE) and Mohs micrographic surgery (MMS). WLE relies on over-estimating the boundary of a tumor to completely remove it in a single surgical pass. While this can be effective for tumors with well-defined margins, many BCCs are more complex with unpredictable margins [27]. WLE can result in recurrence rates as high as 50% over 10 years if a tumor is not entirely excised [28].

MMS was first described in 1941 by Dr. Frederic Mohs and is used to treat many skin malignancies [29,30]. The surgery involves horizontally shaving a lesion in thin sections and evaluating each section by microscopy to detect cancerous tissue. Further excision is performed only where the tumor remains detected [31]. It promotes the identification of complete tumor margins while minimizing non-diseased tissue removal [32]. For BCC specifically, Mohs surgery is more effective than WLE in preventing tumor recurrence in both primary and recurrent lesions [30,33]. MMS is recommended for BCCs that exhibit more aggressive behavior and are situated at cosmetically sensitive locations to reduce disfiguration of the patient [34]. However, Mohs surgery requires highly skilled and trained surgeons on this complex technique and may be less available to patients in underdeveloped regions [28]. Additionally, some patients may refuse surgical intervention for personal reasons or are medically excluded as candidates. Of primary concern are elderly patients who may not properly recover from surgery and the potential cosmetic consequences of such invasive surgical procedures.

### 2.2. Radiation Therapy

BCC patients that cannot undergo or refuse surgical treatment for BCC require other therapeutic options. For example, lesions on the eyelids, nose, lips, and ears can be extremely challenging to completely excise surgically. Attempted excision may result in compromised anatomical function alongside undesired cosmetic outcomes [35]. Radiation therapy has emerged as one alternative in these situations, primarily due to its ability to treat both low- and higher-risk tumors [14]. Overall, the recurrence rate following radiation therapy alone is less than 10% over 5 years [36]. Radiotherapy is also helpful for cooperative treatment following surgery when margins are poorly defined or excision is incomplete [37].

Radiotherapy does, however, have limitations that are necessary to consider. Efficacy is dependent on tumor size and should be reserved for smaller tumors [28]. While older patients that are not surgical candidates can benefit significantly from radiotherapy, younger patients may experience an undesired cosmetic decline over time [36]. Any treatment with ionizing radiation increases the likelihood of developing other cancers, including melanoma. The destruction of tumor tissue cannot be guaranteed or determined without a more invasive follow-up. Most importantly, radiotherapy is rarely recommended for patients with GS [38]. Treatment with radiotherapy, even for non-skin cancer GS symptoms, may induce rampant BCC growth [39]. The complete destruction of tumor tissue cannot be guaranteed or determined without a more invasive follow-up.

### 2.3. Laser Therapies

Low-risk nodular and superficial BCCs may be non-invasively treated using laser-based techniques. Ablative therapy can be performed by carbon dioxide (CO_2_) or doped yttrium aluminum garnet (YAG) lasers [40]. Recurrence rates for this type of therapy are remarkably low at less than 3% [41]. Patients generally report favorable cosmetic outcomes compared to surgical resection [42], though it is unclear if this is due to inherent differences in tumor type. Mohs surgery is rarely used for smaller, lower-risk nodular/superficial BCCs, but laser treatments are. Scarring is logically expected to be less extreme for treatment of a smaller tumor.

Another laser-based technique for BCC treatment is photodynamic therapy (PDT). The mechanism of PDT is well characterized and, truthfully, most beneficial for treating skin cancers. A photosensitizer is delivered either topically or systemically to a tumor and irradiated with visible light that matches the excitation frequency of the sensitizer. The excited state transfers energy to water, producing cytotoxic singlet oxygen (^1^O_2_) and other reactive oxygen species (ROS) (Figure 2) [28]. Due to the limited depth penetration of visible light, PDT has struggled to gain clinical approval to treat most cancers. However, PDT is an approved treatment for BCC in 18 countries [43,44]. In particular, superficial BCC has been extensively studied as a model cancer for evaluating PDT efficacy [44].

The safety profile of PDT is ideal for a tumor with such a high recurrence rate. The laser light is not harmful to the genetic composition of healthy tissue and is locally administered with high precision to the sensitizer-bearing tumor. PDT, however, carries some of the highest variability in treatment efficacy, with cure rates between 50–90% for primary tumors and as low as a 20% cure rate for recurrent tumors [16]. PDT can cause severe sensitivity to the sun post-treatment and may take several rounds to be maximally effective [45]. In contrast to radiotherapy, PDT is safe for patients with GS and effective in lesions with less than 2 mm depth (deeper tumors are not treatable by PDT due to limited tissue penetration of laser light) [46,47]. Additional research and optimization are required to improve the general efficacy of PDT against BCC.

### 2.4. Imiquimod and 5-Fluorouracil Topical Treatments

Imiquimod (Aldara; 3M Pharmaceuticals, Figure 1, left and 5-fluorouracil (Adrucil^®^, 5-FU, Figure 1, right are topical creams applied to BCC lesions for chemotherapeutic treatment. Imiquimod is approved by the Food and Drug Administration (FDA) for treating superficial BCCs less than 2 cm in diameter and is being evaluated for efficacy in nodular BCCs [48]. Therapeutic effect is achieved through an immune response and induction of apoptosis [49]. 5-FU is approved by the FDA for the treatment of superficial BCCs. The inhibition of nucleic acid synthesis is the primary mechanism of action. One study reported a 90% cure rate, with patients experiencing minimal side effects [50]. In both therapies, intense skin reactions are observed due to inflammation caused by imiquimod and lack of tumor specificity of 5-FU [28,51,52]. However, both therapies were found to be most effective when used in conjunction with other treatment options, suggesting that their utility might be optimized synergistically [51,53]. Topical treatment is ideal for skin cancers- localized delivery minimizes the harm to healthy tissues, specifically parts of the body completely unaffected by cancer. However, superficial BCC represents only 10–30% of the BCC patient population, severely limiting the number of patients who can access these treatments.

## 3. The Hedgehog Signaling Cascade in BCC

In the 1990s, genetic evaluation of BCCs of patients with GS revealed the most crucial discovery in BCC research history: BCC lesions are often linked to mutations in the patched1 (*PTCH1)* gene loci [54,55,56,57]. Since then, it has become commonly accepted that the Hedgehog (Hh) signaling cascade, to which PTCH1 proteins belong, is BCC’s primary oncogenic driver [58,59,60]. The Hh pathway is canonically activated (Figure 2) by the binding of Hh ligands to the transmembrane protein PTCH1, which releases smoothened (SMO) inhibition. Suppressor of fused (SUFU) is signaled to release glioma-associated oncogene (Gli) transcription factors where they are activated in the cytosol. Translocation into the nucleus activates the expression of target genes for cellular processes such as proliferation and migration [61,62]. Dysregulation of this pathway is associated with many cancers but is causative of BCC [58,59,60,63,64]. As such, it is a promising chemotherapeutic target for BCC.

Approximately 90% of sporadic BCCs arise from mutations of one *PTCH1* allele, and 10% harbor mutations to downstream protein SMO [65]. Mutations in tumor suppressor p53 (p53) are also observed in BCC [66]. These mutations are consistent with genetic modifications commonly caused by UV exposure that ultimately leads to increased proliferation, maintained stemness, and tumorigenesis [23,65,67,68,69]. Additionally, activation of Hh signaling is often associated with the overexpression of programmed cell death ligand (PD-L1), promoting immunogenic escape and tumor cell proliferation [70,71].

## 4. Chemotherapies That Target Hedgehog Signaling

While treatment for lower-risk BCCs is mainly successful across the treatments described above, patients who suffer from high-risk infiltrative BCCs have fewer treatment options [72]. Infiltrative BCCs are broken further into locally advanced BCCs (laBCC) and metastatic BCCs (mBCC), and often, surgical resection is not an option for these patients. While mBCC only occurs in less than 0.5% of cases [72,73], it presents a unique treatment challenge. The majority of metastasis is observed in the lymph nodes, lungs, liver, and bone [72]. Before the development of systemic chemotherapies targeting Hh signaling, median survivability for mBCC patients was only 8 months after diagnosis [74].

### 4.1. Smoothened Inhibitors

The overwhelming majority of Hh-specific therapies target the transmembrane protein SMO. The first described SMO antagonist is Cyclopamine (Figure 3A), a natural product found in corn lily [75]. Pregnant ewes grazed on corn lily produced offspring with craniospinal defects, including cyclopia, that could not be explained [76,77]. During this time, a connection between mutations in the Hh pathway genome and the occurrence of holoprosencephaly (including cyclopia) in mammals was found [78,79]. After extensive research, inhibition of SMO by what is now commonly referred to as Cyclopamine was determined to cause birth defects in the ewe litters [80,81]. While studied extensively as a chemotherapeutic agent [82,83,84,85,86,87,88], Cyclopamine suffers from poor bioavailability due to a lack of solubility and stability [75]. However, the structural elucidation of Cyclopamine promoted the development of analogs with improved biocompatibility.

Two SMO inhibitors have received FDA approval for the treatment of advanced BCCs. Vismodegib (GDC-0449, Erivedge^®^, Figure 3B) is approved to treat recurrent, locally advanced, and metastatic BCCs in patients who are not candidates for surgery or radiation therapy [89]. Sonidegib (Odomzo, Novartis, Figure 3C) is approved for laBCC in patients who are not candidates for surgery or radiation therapy [90]. Both inhibitors have shown efficacy for some patients whose outcomes might have otherwise been poor, but they certainly are a far cry from the perfect answer to BCC treatment. Approximately 50% of patients treated with Vismodegib have no initial response, and of those that do, over 20% develop chemo-resistant tumor recurrence [91,92]. Many of the mutations that lead to chemoresistance were identified in the drug target, SMO, suggesting that mutations of SMO structure in primary tumors may explain the lack of response experienced by some patients [93,94,95]. Chemoresistant tumor recurrence is a significant issue considering that BCC already exhibits such high recurrence rates. Vismodegib is administered orally as this is the only way to ensure that metastases are effectively treated but means that all areas of the body are exposed to the drug. Secondary BCC lesions that could grow in new locations might also develop resistance to further Vismodegib treatment. Additionally, patients receiving these treatments often experience untoward side effects such as muscle cramps, loss of taste, weight loss, hair loss, and mental health decline that are not always amenable to continued therapy [96,97]. Muscle cramps are most common due to the activation of calcium flux upon inhibition of canonical Hh signaling. Other symptoms such as hair and taste loss stem from the systemic inhibition of Hh signaling required for the maintenance of hair follicles and taste buds. Approximately 20% of BCC patients enrolled in various trials with SMO inhibitors discontinue treatment due to these side effects [97].

**Figure 3 biomedicines-10-02376-f003:**
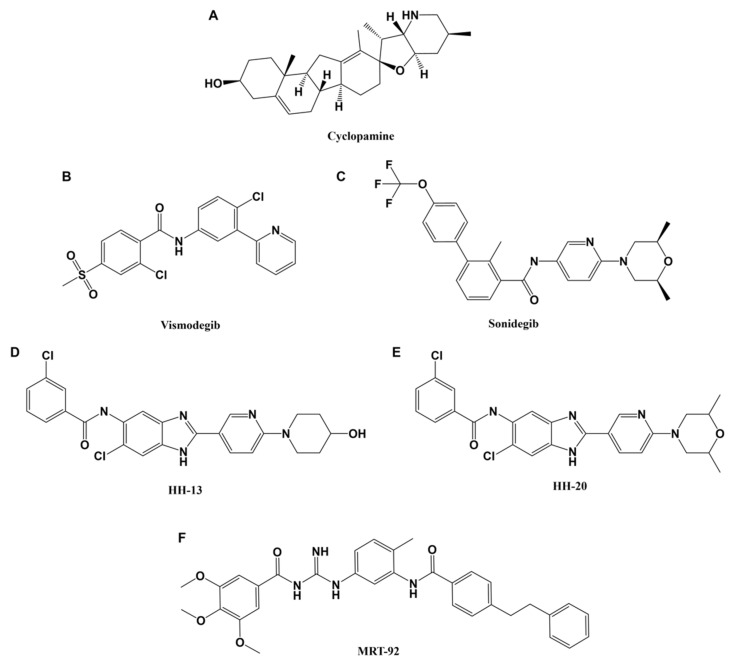
Chemical structures of smoothened inhibitors (**A**) Cyclopamine. (**B**) Vismodegib. (**C**) Sonidegib. While distantly related to Cyclopamine, Vismodegib and Sonidegib contain structural similarities. All three inhibitors bind SMO in the same pocket. (**D**,**E**) New generation Vismodegib derivatives developed by Li et al. in 2019 [98]. (**F**) To date, the most potent SMO inhibitor reported by Hoch, et al.

However, chemoresistant recurrence remains a serious concern. In one study, researchers attempted to treat Vismodegib resistant tumors with Sonidegib to test the hypothesis that a different SMO inhibitor might still be effective. The results of this study concluded that patients with Vismodegib resistance are likely for their disease to continue to progress if treated with Sonidegib, specifically [96]. A limitation of this study is that Vismodegib and Sonidegib have similar chemical structures and bind SMO in the same location [96]. It cannot be concluded that all SMO inhibitors would be ineffective, but only that Sonidegib was insufficient to overcome Vismodegib resistance.

Recent studies have expanded upon structural components of Vismodegib that have resulted in more effective therapeutics and attenuation of resistance [98,99]. One study identified two new molecules as potent SMO inhibitors that are loosely founded upon the structure of Vismodegib. These compounds labeled HH-13 (Figure 3D) and HH-20 (Figure 3E) displayed 10 and 30 nM IC_50_ values in cellular assays of Hh activity, respectively [98]. Most importantly, these compounds remained effective against SMO-D473H, a SMO mutant that Vismodegib is incapable of inhibiting. Vismodegib efficacy is diminished by almost 1000-fold between wild type SMO and SMO-D473H, whereas the effectiveness of HH-13 and HH-20 is only decreased by 1.1 and 1.4-fold, respectively. While this is an exciting advancement in the realm of development of SMO inhibitors, only one mutant version of SMO was evaluated. It is likely that HH-13 and HH-20 will not be effective against all SMO mutants.

To date, the most effective SMO inhibitor was first described by Hoch et al. in 2015 and inhibits SMO with approximately 10-fold improvement in potency over Vismodegib. MRT-92 (Figure 3F) is an acylguanidine derivative with structural differences from Vismodegib that promote binding to a different locale in the SMO structure. Whereas Vismodegib binds SMO in the extracellular domain, the MRT-92 scaffold was shown to bind the entire length of the SMO transmembrane domain. This provides a competitive advantage against common SMO mutations. MRT-92 remains potent against the SMO-D473H mutation due to retained binding affinity where Vismodegib binding affinity is lost entirely [100]. Additionally, MRT-92 successfully controlled the tumor growth of a murine xenograft melanoma model, suggesting applicability to BCC [101].

Another possible solution to improve SMO inhibition is the exploration of chemical structures that deviate from that of Vismodegib. In 2015, the Waldmann group identified that synthetic modifications to the natural product withaferin A (Figure 4A) produced potent inhibitors of SMO. Specifically, compound 21a (Figure 4B) exhibited a strong binding affinity for SMO and an IC_50_ around 2 µM [102]. However, the synthesis of these complexes is non-trivial and is stereoselective. Diastereomerization of 21a to 21a’ (Figure 4C) reduced potency by almost 5-fold [102].

In 2021, the Passarella group proposed simplifying the structure to contain cyclic carbamates with the ultimate goal of synthesizing and evaluating compound 1 (Figure 4D) [103]. However, stereoisomers of this compound ultimately proved inactive against Hh signaling. Two pathway intermediates, 13b (Figure 4E) and 14b (Figure 4F), successfully inhibited SMO with racemic IC_50_ values of 7.4 µM and 13.0 µM, respectively. Enantioselective synthesis revealed that (+)-13b and (−)-14b were the more potent inhibitors with IC_50_ values around 6 µM compared to their enantiomers at 11–16 µM [103]. While it is surprising that deprotection of 14b to yield compound 1 eliminates activity, the tert-butyl(chloro)diphenylsilane (TBDPS) protecting group is highly lipophilic and might significantly impact protein interaction. While the potency of these complexes does not compare to clinically available Vismodegib and Sonidegib, further study is necessary to evaluate inhibition in BCC specifically and the ability to evade resistance.

### 4.2. Gli Inhibitors

While newer generations of SMO inhibitors with structurally diverse scaffolds are promising, it is unclear if they will successfully evade the complications of the already approved compounds. As mentioned above, mutations of SMO itself render continued treatment with SMO inhibitors ineffective if the mutation abolishes drug binding. Additionally, cellular switches have been identified to bypass SMO activity in some recurrent tumor pathology [104]. Therefore, options targeting other Hh signaling regulators would be beneficial for patients who do not initially respond to treatment with SMO inhibitors or develop resistance. In one study addressing Vismodegib resistance, mutations of SMO proteins were the primary focus. Two smoothened variants known to be insensitive to Vismodegib were expressed in *SMO* knockout mouse embryonic fibroblasts. Upon treatment with both direct and indirect Gli transcription factor inhibitors, Hh activity was indeed reduced regardless of the identity of the SMO mutation [94]. In another study, activation of serum response factor and the transcriptional cofactor megakaryoblastic leukemia 1/2 (MKL1/2) were found to have a novel, non-canonical interaction with Gli1 that amplified Hh transcription independently of SMO. Excellent in vivo anticancer activity was achieved through MKL1/2 inhibition in Vismodegib resistant tumors with this characteristic [105]. Interestingly, both the canonical and non-conical resistance mechanisms ultimately influence the activity of Gli1 transcription factors in a way that is druggable [106]. As such, Gli is a valuable target for chemotherapeutic intervention in BCC.

To date, no therapies targeting the Gli family of zinc finger transcription factors (ZnFtfs) have received clinical approval. This is primarily attributed to the fact that transcription factors, in general, are notoriously challenging to target specifically with traditional small molecules due to a lack of well-defined binding pockets [107,108,109]. However, a few small molecule inhibitors for Gli proteins have been developed and studied against Hh signaling. Figure 5 depicts the inhibitors discussed in this section.

One of the first small molecules found to inhibit Hh signaling through Gli downregulation is arsenic trioxide (ATO, Figure 5A) [110]. Importantly, ATO is effective in the treatment of tumors that have developed SMO resistance around a dose of 500 nM [111]. However, ATO is not specific to Gli transcription factors and is known to bind numerous intracellular targets [112]. In fact, it is FDA approved (Trisenox, Cell Therapeutics) for the treatment of acute myeloid leukemia due to its ability to potently inhibit promyelogenous leukemia-retinoic acid receptor fusion protein [113,114]. Ideally, a Gli inhibitor would be both potent and specific to reduce unwanted off-target complications.

The most prominent small molecule Gli inhibitor is GANT-61, a derivative of the Gli antagonist (GANT) family of compounds. It was discovered in 2007 and has since been used to study Hh inhibition in a variety of cancers [115,116,117,118,119,120,121,122]. The general inhibitory concentration at which 50% of Hh signaling is reduced (IC_50_) is on the order of 5–10 µM for GANT-61 [123,124]. GANT-61 is understood to undergo a prodrug mechanism where hydrolysis of the intact molecule produces an inactive side product and the active inhibitor (Figure 5B) [125]. Computational analysis further suggests a direct binding mechanism of GANT-61 to Gli transcription factors that inhibits DNA binding and therefore transcription, but this has yet to be confirmed experimentally [125,126,127].

Ultimately, GANT-61 is limited by poor solubility and bioavailability [128]. More recently, the natural product Glabrescione B (GlaB, Figure 5C) was reported as the first confirmed small molecule to directly bind Gli and prevent the Gli/DNA binding interaction. GlaB inhibited BCC growth in vitro and in vivo at the equivalent of low µM doses [129]. When compared directly to GANT-61 in this work, GlaB was found to have no significant improvement in potency or inhibitory effect. However, GANT-61 efficacy was only directly compared with in vitro experiments, not in vivo. It is possible that GlaB exhibits higher bioavailability and would be more effective in vivo. However, a study of GlaB against Hh activity in medulloblastoma showed that micelle encapsulation improved solubility and potency, revealing that GlaB efficacy similarly suffers from low bioavailability [130].

The discovery of Gli inhibition by natural product GlaB suggests potential core chemical structures that could by synthetically modified to improve solubility and binding to Gli transcription factors. Specifically, modifications to the isoflavone core (Figure 5C,D) have been made to study the effect of structure on efficacy. Chemical modifications at the meta and para positions of the third ring generate compounds that influence inhibitory potential. Compounds 5 and 12 (Figure 5D) inhibit Gli with IC_50_ values of in the 2–10 µM range, similar to GlaB [131]. However, compound 17 did not show inhibitory potential under 30 µM, suggesting that very small structural changes make large differences in protein binding.

Further chemical modification of this scaffold has revealed that bulky substituents at the *meta* position produce isoflavones that target Gli, but bulky substituents at the *para* position generate compounds that target SMO [131,132]. Combining these principles yielded compound 22 (Figure 5E) which targets both SMO and Gli. This agent successfully inhibited tumor growth in a model of medulloblastoma. However, the ability to target two proteins means that selectivity for this compound is questionable. It is unknown what other molecular targets it may bind, producing unwanted off-target effects.

In 2020, a new structural scaffold was reported to inhibit Gli transcription factors. Bicyclic imidazolium compounds were first discovered to inhibit Gli in a high throughout drug screen and then were evaluated across a range of structure/activity relationship studies [133]. Of 12 synthesized molecules, compound 10 (Figure 5F) was found to have the highest anti-Gli activity with an IC_50_ between 100 nM and 5 µM, depending on the assay. Structural considerations determined that the terminal phenyl group added significant potency to the molecules, whereas adding heteroatoms into the bi/tricyclic ring system essentially eliminated anti-Gli activity [133]. It is worthy to note that these complexes are highly aromatic and hydrophobic, much like GANT-61 and GlaB. It is likely that solubility will continue to be an issue for these compounds, though this was not discussed by the authors. Additionally, these complexes significantly interfered with mitochondrial health and function, something that must be considered within the context of selectivity and unwanted side effects in non-diseased tissue.

The discovery of cisplatin as an anticancer agent birthed an entirely new field of medicinal inorganic chemistry [134,135]. Inorganic compounds can be desirous as therapeutics for their ability to access inhibitory mechanisms unknown to organic compounds. In recent studies, the zinc ions that structurally support the alpha-helical structure of Gli transcription factors were targeted by a cobalt-Schiff base inorganic complex (Figure 6A) [136,137]. Cobalt-Schiff base complexes have been shown to displace the zinc ions from Cis_2_His_2_ coordination packets via preferential histidine binding of cobalt [138]. The alpha-helical structure of Gli is then depleted in the DNA binding domain. As a result, the Gli/DNA interaction is inhibited, and target genes would not be transcribed [137].

To achieve specificity, the consensus DNA sequence that only one transcription factor will recognize is tethered to the active cobalt-Schiff base inhibitor. This brings the cobalt complex into close enough proximity to the protein to irreversibly displace zinc ions. This strategy has been previously employed in *Drosophila* and *Xenopus* model organisms of homologous Hh pathways [139,140,141]. CoGli (Figure 6B) was developed by Dukes et al. to inhibit the Gli family of transcription factors in cellular assays of BCC. In this study, GANT-61 served as a positive control. GANT-61 inhibited Hh-driven migration of the ASZ murine BCC cell line by approximately 50% at a 5 µM dose. Strikingly, the targeted cobalt-DNA complex delivered by a cationic vehicle inhibited Hh-driven migration by 50% at only 300 nM [136]. This represents a promising new direction in the field of Gli inhibition as the first inhibitor to exhibit nM efficacy with a high degree of selectivity.

While GANT-61 has seemingly exhausted its potential for clinical translation, GlaB derivatives and imidazolium compounds show potential for further development as potent organic Gli inhibitors. Cobalt-DNA based complexes show additional promise with unique inhibition mechanisms and improved target selectivity. However, few of these compounds have been thoroughly evaluated for Hh inhibition in BCC, specifically. It is challenging to conclude their applicability to BCC treatment but highlights the need for further study and development for an effective Gli inhibitor to achieve clinical approval.

## 5. Preclinical Models for Hedgehog and BCC Research

A significant challenge for the BCC research community is the overall paucity of preclinical models for evaluating therapeutics. This is even more problematic when considering the unavailability of human-derived models. While animal models are broadly used as research tools in therapeutic development, fundamental differences between different species directly impact the translation of therapeutics. Here, we outline the current and recent therapeutic evaluation developments in both Hh activity cell lines and BCC-specific model systems.

### 5.1. Hedgehog Activation in Cellular Assays

The first benchmark for evaluating a new Hh inhibitor is often a cellular assay of Hh activity. One example is a luciferase reporter assay performed in a derivative of the NIH-3T3 mouse fibroblast cell line that contains modified Gli binding domains driving expression of firefly luciferase [142]. When Hh signaling is exogenously activated, luciferase is stably expressed and can be measurably down-regulated by concurrent treatment with Hh inhibitors [143,144,145]. The pathway can be activated at PTCH1, SMO, or by transfecting a plasmid encoding for Gli transcription factors to model Gli accumulation.

The C_3_H/10_T_1/2 cell line is a pluripotent mouse embryonic fibroblast routinely used in Hh research [146]. The cells do not exhibit innate Hh activity, but exogenous activation promotes differentiation into osteoblasts and induces alkaline phosphatase (ALP) protein expression [147,148,149]. Concurrent treatment with Hh inhibitors results in measurable prevention of ALP production. This cellular system has dramatically increased the understanding of basic Hh mechanics and the general efficacy of inhibition strategies [150]. However, both NIH-3T3 and C_3_H/10_T_1/2 cellular assays are not representative of Hh dysregulation in a tumor environment and are also not derived from skin cells. While suitable for initial evaluation of Hh inhibitory potential, more specific models are necessary to evaluate applicability to BCC.

### 5.2. Murine BCC Models

Rodents (mice and rats) are choice mammals for most early-stage preclinical investigations of cancer treatments. Developing a rodent model that most closely mimics human disease is essential for successful translation into the clinic. As such, it is desirous for an animal to grow tumors spontaneously. One of the first mouse models to spontaneously developed BCC was generated by overexpressing sonic hedgehog (SHH) proteins that initiate Hh signaling [151]. SHH is a paracrine signal, however, and activation was not isolated to skin cells malformations across the animal were observed.

Additionally, the animals had to be examined either in the embryonic or neonatal states due to uncontrollable perinatal lethality. While embryos did develop large BCC-like lesions that mimicked patient BCC phenotype and pathology, they most closely resembled uncontrollable GS BCC growth. Animals allowed to grow long enough to die in utero had large sections of skin destroyed from advanced disease [151]. Ultimately, the untimely death of the animals in this study prevents the development of a breedable line that could be used to investigate Hh inhibitors. However, it provides evidence for the ability to develop murine BCC models through manipulation of the Hh genome.

Other studies attempted a similar generation of spontaneous BCC models through transgenic Hh activation. One report induced expression of a constituently active mutant of SMO under a keratin 5 promoter that confined expression to the skin. BCCs developed in embryos that mimicked patient phenotype and pathology [152]. The authors do not comment on perinatal lethality, but further research determined that animals that do survive cannot reproduce to generate a breedable line [153]. However, localized expression in the skin avoided the craniospinal defects seen in SHH overexpression, reducing discomfort and suffering of surviving animals.

*PTCH1* alleles are the most common source of mutations leading to sporadic BCCs in humans. As such, knockdown of *Ptch1* has been attempted for the development of spontaneous BCCs in mice. An extensive review of *Ptch1* knockout mice has been previously published [154]. Here, we focus on the broad story of development.

Early efforts towards this aim proved fruitless. Animals were viable but developed medulloblastomas [155] and rhabdomyosarcomas [156] and even other symptoms of GS but did not produce apparent BCCs. Further studies revealed that *Ptch1* heterozygous mice at 9 months had small proliferations of BCC-like cells that could only be detected microscopically [153]. To encourage tumorigenesis, mice were subjected to UV or X-ray irradiation. After UV irradiation, *Ptch1* knockout mice had a 20% incidence of developing lesions that mimicked the phenotype of human BCCs. X-ray irradiation produced trichoblastomas primarily.

The generation of BCCs from *Ptch1* knockout mouse models resulted in an equally important development: BCC’s first immortalized cell lines. Three cell lines were isolated and immortalized from three different mouse models. The most commonly studied ASZ001, or ASZ, the cell line was immortalized from a BCC lesion resulting from UV irradiation three times weekly for 10 months [153]. These cells retain knockdown of *Ptch1* in culture and are verified to be sensitive to Hh inhibition.

When irradiated, tumors on *Ptch1* knockdown mice develop in a controlled manner where the UV light is applied. This significantly reduces the number of lesions from a truly spontaneous model and allows for a more controlled experimental design. Notably, some mice developed tumors in as little as four months of UV irradiation [153]. After histopathological validation of tumor type, these lesions could be treated with Hh inhibitors on a semi-reasonable time scale. However, the model produces a significant time burden from breeding to birth to tumor development. The answer to this rime delay for many cancers is patient-derived xenograft (PDX) models. Cells from human cancer patients are injected into an immunocompromised animal and develop into a tumor [157]. The time scale for this is exponentially faster and does not require the use of heavily genetically modified animals. Additionally, cancer is now fundamentally of human identity.

Unfortunately, developing PDX models of BCC has largely failed. Tumors often do not implant or are met with slow growth rates [158,159]. The use of more severally immunocompromised mice improves implantation [160,161], but little is known about the retention of cellular identity and behavior to the original tumor [162]. However, one study successfully allografted murine BCC cells into an immunocompromised mouse model with the assistance of Matrigel. Allografted tumors retained the phenotype and pathology of their parent tumors and were responsive to inhibition. Most importantly, allografts produced visible tumors within only 3 weeks of implantation [162]. This technique is unique and could be applied to the grafting of patient samples for a more human-based model of BCC.

As mentioned previously, there is an inherent distant relationship between the tissues of rodents and humans. Therefore, without available PDX models, a species more closely related to human identity is a valuable research tool. In 2017, a group investigated the ability to generate a BCC model in a non-human primate, the Chinese tree shrew [163]. Chinese tree shrews are small and have been used to study many human diseases [164]. Their skin is anatomically similar to human skin, creating a unique opportunity for more accurate BCC model development [165]. Development of the tree shrew model of BCC was accomplished via lentiviral transfection of SmoA1, a constitutively activated form of SMO. Injections were performed in both dorsal and tail skin, resulting in the development of BCC lesions that mimic human BCC. While this model has yet to be used to investigate BCC inhibition, it provides an interesting preclinical link between murine and human species.

## 6. Perspectives

While many BCC cases are readily cured via surgical methods, both ineligibility and personal choice may leave many BCC patients without effective treatment options. As this cancer affects millions of people worldwide, alternative solutions are desirable. Herein, we have described scientific advancements in the development of targeted therapies for SMO and Gli proteins involved in the Hedgehog signaling cascade. Hh inhibitors are desirous due to the causative relationship between dysregulated signaling and BCC oncogenesis and progression. However, very few inhibitors have successfully translated to the clinic.

The two Hh targeted therapies that have reached clinical approval target SMO, an upstream Hh regulator prone to chemo-resistant mutation. Additionally, cellular switches and non-canonical Gli activation often render long-term SMO treatment ineffective. Gli has been identified as an alternative target but is challenging to specifically inhibit as few chemical structures selectively interact with Gli. While developing new chemical structures recognized to bind Gli and novel inorganic inhibition strategies are encouraging, many aspects of drug bioavailability must be optimized before these agents can be translated to the clinic.

Upon considering the available in vitro and in vivo models of BCC, we also identify this area of the research field to be lacking. The growth of spontaneous BCC tumors is very slow in successful transgenic mouse models, significantly hindering the time in which a research study can be performed. Additionally, many of these systems require the exposure of the animals to UV irradiation that mimics a moderate sunburn multiple times a week for several months. For these experiments to be justified and fruitful, we identify the need for an in vitro assay that is more complex than simple 2-dimensional cellular assays and mimics the characteristics of BCC in tissue. For many cancers, this can be accomplished through the development of 3D spheroid cultures. For skin cancer specifically, researchers have succeeded in developing 3D epidermal mimics that can be assembled to mirror a skin cancer of interest [166]. We propose the development of a similar model of BCC to be of high utility to the research field at large.

Finally, the described preclinical models of BCC have only been used to evaluate the treatment of established disease. They have not been utilized to study how BCC latent potential ultimately leads to lesions. This should be a significant research endeavor considering BCC’s rate of primary and secondary tumor recurrence. Understanding these mechanisms in skin cells might elucidate new ways BCC can be prevented. All BCC patients should be considered at high risk for developing multiple BCCs, and effective prevention strategies would significantly improve the lives of BCC patients.

## Data Availability

Not applicable.

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
