# Peer review of "Modulation of Hedgehog Signaling for the Treatment of Basal Cell Carcinoma and the Development of Preclinical Models"

_biomedicines, 2022, doi:10.3390/biomedicines10102376_

Round 1
Reviewer 1 Report
This review provides an overview of the state of knowledge in the field of basal cell carcinoma research with emphasis on the therapeutic implication of hedgehog pathways.
There were several things that the authors did not touch base on but the readers might be interested in, such as the impact of hedgehog signaling on the tumor microenvironment of BCC, the link between hedgehog signaling and PD-1/PD-L1 expression, mechanisms of tumor resistance to hedgehog-targeted therapies. Information on those areas will offer a deeper appreciation of the significance of hedgehog signaling pathway in BCC.
Author Response
This review provides an overview of the state of knowledge in the field of basal cell carcinoma research with emphasis on the therapeutic implication of hedgehog pathways.
There were several things that the authors did not touch base on but the readers might be interested in, such as the impact of hedgehog signaling on the tumor microenvironment of BCC, the link between hedgehog signaling and PD-1/PD-L1 expression, mechanisms of tumor resistance to hedgehog-targeted therapies. Information on those areas will offer a deeper appreciation of the significance of hedgehog signaling pathway in BCC.
We appreciate the suggestions from reviewer 2 will add depth to the review material. We have added discussion of the possible role of PD-1/PD-L1 in tumor persistence and proliferation. While we mentioned the role of SMO mutations in resistance to Hh-targeted therapies, we expanded upon this further and also mentioned other resistance mechanisms that bypass SMO signaling altogether.
While the tumor microenvironment in BCC might be of interest to some readers, we find in-depth discussion of this to be outside the scope of this review and hope the reviewer would understand our desire to stay focused on therapies that target proteins directly involved in the Hh cascade.
Reviewer 2 Report
In this review, Dukes and Meade summarize the current state of research in BCC treatment, focusing on Hedgehog (Hh) signaling cascade, the molecular driver of BCC. They provide an in-depth review of the molecules/drugs that target the Hh pathway at the level of the main transducer Smoothened and the downstream GLI transcription factors. They also discuss mouse models of BCC that could be used in pre-clinical research.
Overall, the review covers most of all the literature regarding the focus of the subject and is a valuable addition to a growing list of reviews on this and related topics. The role of Hh pathway in BCC is an import topic in the cancer field and readers (both basic scientists and clinicians) who are interested in this field will greatly benefit.
I only have few suggestions that may help improving the manuscript.
1) could the authors provide an explanation why SMO antagonists cause so many adverse events (page 9, lines 260-262)?
2) address the point about Gorlin syndrome and risk of specific cancers. In particular, discuss why patients with Gorlin syndrome are predisposed to BCC, but they are not at risk to develop other Hedgehog dependent cancers.
3) In the paragraph regarding SMO inhibitors, the authors did not mention two studies about a new class of SMO inhibitors based on acylguanidine scaffold (MRT-92, also known as EPMF03):
-Hoch L, Faure H, Roudaut H, Schoenfelder A, Mann A, Girard N, Bihannic L, Ayrault O, Petricci E, Taddei M, Rognan D, Ruat M. MRT-92 inhibits Hedgehog signaling by blocking overlapping binding sites in the transmembrane domain of the Smoothened receptor. FASEB J. 2015 May;29(5):1817-29. doi: 10.1096/fj.14-267849.
-Pietrobono S, Santini R, Gagliardi S, Dapporto F, Colecchia D, Chiariello M, Leone C, Valoti M, Manetti F, Petricci E, Taddei M, Stecca B. Targeted inhibition of Hedgehog-GLI signaling by novel acylguanidine derivatives inhibits melanoma cell growth by inducing replication stress and mitotic catastrophe. Cell Death Dis. 2018 Feb 2;9(2):142. doi: 10.1038/s41419-017-0142-0.
These papers show that MRT-92 uniquely binds to the entire transmembrane cavity of SMO and is effective against the human SMO-D473H, a key mutation that renders SMO resistant to Vismodegig or Sonidegib (Hoch et al, FASEB J. 2015). In addition, MRT-92 is among the most potent SMO antagonists, being 10-fold more potent than Vismodegib or Sonidegib (Pietrobono et al, Cell Death Dis. 2018).
Minor points:
-Replace “HH proteins” with “HH ligands” (page 6 and Figure 2 legend).
-At line 427 (legend Fig. 6), replace “phosporothioate” with “phosphorothioate”.
Author Response
In this review, Dukes and Meade summarize the current state of research in BCC treatment, focusing on Hedgehog (Hh) signaling cascade, the molecular driver of BCC. They provide an in depth review of the molecules/drugs that target the Hh pathway at the level of the main transducer Smoothened and the downstream GLI transcription factors. They also discuss mouse models of BCC that could be used in pre-clinical research.
Overall, the review covers most of all the literature regarding the focus of the subject and is a valuable addition to a growing list of reviews on this and related topics. The role of Hh pathway in BCC is an import topic in the cancer field and readers (both basic scientists and clinicians) who are interested in this field will greatly benefit.
I only have few suggestions that may help improving the manuscript.
1) could the authors provide an explanation why SMO antagonists cause so many adverse events (page 9, lines 260-262)?
We have updated this section to include an explanation for muscle cramps, hair loss, and taste loss associated with Vismodegib/Sonidegib treatment. The has been updated and is below:
“Additionally, patients receiving these treatments often experience untoward side effects such as muscle cramps, loss of taste, weight loss, hair loss, and mental health decline that are not always amenable to continued therapy[94,95]. Muscle cramps are most common due to the activation of calcium flux upon inhibition of canonical Hh signaling. Other symptoms such as hair and taste loss stem from the systemic inhibition of Hh signaling required for the maintenance of hair follicles and taste buds. Approximately 20% of BCC patients enrolled in various trials with SMO inhibitors discontinue treatment due to these side effects [95].”
2) address the point about Gorlin syndrome and risk of specific cancers. In particular, discuss why patients with Gorlin syndrome are predisposed to BCC, but they are not at risk to develop other Hedgehog dependent cancers.
We are aware of other symptoms patients with GS suffer from. While BCC is the most common cancer GS patients develop, they also have a propensity for medulloblastoma (MB), a malignant childhood brain cancer. While many cancers in general have Hh dysregulation as a piece of their pathology, BCC and SHH subtype MB are two cancers known to be directly caused by Hedgehog dysregulation. As such, they are the cancers GS patients are most likely to develop. Additionally, patients with GS often suffer from craniospinal defects associated with holoprosencephaly caused by mutations in the Hh genome at birth. We have adjusted the introduction to GS to mention these other afflictions common to GS patients.
“90% of patients with GS experience the uncontrolled growth of multiple BCCs alongside various developmental abnormalities including those associated with holoprosencephaly and malignant medulloblastomas [9,10].
3) In the paragraph regarding SMO inhibitors, the authors did not mention two studies about a new class of SMO inhibitors based on acylguanidine scaffold (MRT-92, also known as EPMF03):
-Hoch L, Faure H, Roudaut H, Schoenfelder A, Mann A, Girard N, Bihannic L, Ayrault O, Petricci E, Taddei M, Rognan D, Ruat M. MRT-92 inhibits Hedgehog signaling by blocking overlapping binding sites in the transmembrane domain of the Smoothened receptor. FASEB J. 2015 May;29(5):1817- 29. doi: 10.1096/fj.14-267849.
-Pietrobono S, Santini R, Gagliardi S, Dapporto F, Colecchia D, Chiariello M, Leone C, Valoti M, Manetti F, Petricci E, Taddei M, Stecca B. Targeted inhibition of Hedgehog-GLI signaling by novel acylguanidine derivatives inhibits melanoma cell growth by inducing replication stress and mitotic catastrophe. Cell Death Dis. 2018 Feb 2;9(2):142. doi: 10.1038/s41419-017-0142-0.
These papers show that MRT-92 uniquely binds to the entire transmembrane cavity of SMO and is effective against the human SMO-D473H, a key mutation that renders SMO resistant to Vismodegig or Sonidegib (Hoch et al, FASEB J. 2015). In addition, MRT-92 is among the most potent SMO antagonists, being 10-fold more potent than Vismodegib or Sonidegib (Pietrobono et al, Cell Death Dis. 2018).
We appreciate the recognition of this oversite and have included a paragraph in section 4.1 to reflect these exciting studies.
“To date, the most effective SMO inhibitor was first described by Hoch et. al. in 2015 and inhibits SMO with approximately 10-fold improvement in potency over Vismodegib. MRT-92 (Figure 3F) is an acylguanidine derivative with structural differences from Vismodegib that promote binding to a different locale in the SMO structure. Whereas Vismodegib binds SMO in the extracellular domain, the MRT-92 scaffold was shown to bind the entire length of the SMO transmembrane domain. This provides a competitive advantage against common SMO mutations. MRT-92 remains potent against the SMO-D473H mutation due to retained binding affinity where Vismodegib binding affinity is lost entirely [98]. Additionally, MRT-92 successfully controlled the tumor growth of a murine xenograft melanoma model, suggesting applicability to BCC[99].”
We included the recommended references and have added Figure 3F to showcase the structure of MRT-92.
Minor points:
-Replace “HH proteins” with “HH ligands” (page 6 and Figure 2 legend). -At line 427 (legend Fig. 6), replace “phosporothioate” with “phosphorothioate”.
We have made the recommended changes in the text.